# Technical note: Inherent benchmark or not? Comparing Nash-Sutcliffe and Kling-Gupta efficiency scores

Wouter J. M. Knoben[1,*], Jim E. Freer[2,3], Ross A. Woods[1,3]

[1]Department of Civil Engineering, University of Bristol, Bristol, BS8 1TR, United Kingdom
[2]School of Geographical Science, University of Bristol, Bristol, BS8 1BF, United Kingdom
[3]Cabot Institute, University of Bristol, Bristol, United Kingdom, BS8 1UJ*Now at University of Saskatchewan Coldwater Laboratory, Canmore, Alberta, Canada

*Correspondence to*: Wouter J. M. Knoben (wouter.knoben@usask.ca)

**Abstract.** A traditional metric used in hydrology to summarize model performance is the Nash-Sutcliffe Efficiency (NSE). Increasingly an alternative metric, the Kling-Gupta Efficiency (KGE), is used instead. When NSE is used, NSE = 0 corresponds to using the mean flow as a benchmark predictor. The same reasoning is applied in various studies that use KGE as a metric: negative KGE values are viewed as bad model performance and only positive values are seen as good model performance. Here we show that using the mean flow as a predictor does not result in KGE = 0, but instead KGE = 1-√2 ≈ -0.41. Thus, KGE values greater than -0.41 indicate that a model improves upon the mean flow benchmark – even if the model's KGE value is negative. NSE and KGE values cannot be directly compared, because their relationship is non-unique and depends in part on the coefficient of variation of the observed time series. Therefore, modellers who use the KGE metric should not let their understanding of NSE values guide them in interpreting KGE values and instead develop new understanding based on the constitutive parts of the KGE metric and the explicit use of benchmark values to compare KGE scores against. More generally, a strong case can be made for moving away from ad-hoc use of aggregated efficiency metrics and towards a framework based on purpose-dependent evaluation metrics and benchmarks that allows for more robust model adequacy assessment.

## 1 Introduction

Model performance criteria are often used during calibration and evaluation of hydrological models, to express in a single number the similarity between observed and simulated discharge (Gupta et al., 2009). Traditionally, the Nash-Sutcliffe Efficiency (NSE, Nash and Sutcliffe, 1970) is an often-used metric, in part because it normalises model performance into an interpretable scale (Eq. (1)):

$$NSE = 1 - \frac{\sum_{t=1}^{t=T}(Q_{sim}(t) - Q_{obs}(t))^2}{\sum_{t=1}^{t=T}(Q_{obs}(t) - \overline{Q_{obs}})^2} , \qquad (1)$$

where $T$ is the total number of time steps, $Q_{sim}(t)$ the simulated discharge at time $t$, $Q_{obs}(t)$ the observed discharge at time $t$, and $\overline{Q_{obs}}$ the mean observed discharge. NSE = 1 indicates perfect correspondence between simulations and observations; NSE = 0 indicates that the model simulations have the same explanatory power as the mean of the observations; and NSE < 0 indicates that the model is a worse predictor than the mean of the observations (e.g. Schaefli and Gupta, 2007). NSE = 0 is regularly used as a benchmark to distinguish 'good' and 'bad' models (e.g. Houska et al., 2014; Moriasi et al., 2007; Schaefli and Gupta, 2007), albeit this threshold could be considered a low level of predictive skill (that is, it requires little understanding of the ongoing hydrologic processes to produce this benchmark); it is not an equally representative benchmark for different flow regimes (for example, the mean is not representative of very seasonal regimes but it is a good approximation of regimes without a strong seasonal component (Schaefli and Gupta, 2007)); and it is also a relatively arbitrary choice (for example, Moriasi et al., 2007, define several different NSE thresholds for different qualitative levels of model performance) that can influence the resultant prediction uncertainty bounds (see e.g. Freer et al., 1996). However, using such a benchmark provides context for assessing model performance (Schaefli and Gupta, 2007).

The Kling-Gupta Efficiency (KGE, Eq. (2), Gupta et al., 2009) is based on a decomposition of NSE into its constitutive components (correlation, variability bias and mean bias), addresses several perceived shortcomings in NSE (although there are still opportunities to improve the KGE metric and to explore alternative ways to quantify model performance) and is increasingly used for model calibration and evaluation:

5 $$KGE = 1 - \sqrt{(r-1)^2 + (\alpha - 1)^2 + (\beta - 1)^2} \,, \qquad (2)$$

where $r$ is the linear correlation between observations and simulations, $\alpha$ a measure of the flow variability error, and $\beta$ a bias term (Eq. (3)):

$$KGE = 1 - \sqrt{(r-1)^2 + \left(\frac{\sigma_{sim}}{\sigma_{obs}} - 1\right)^2 + \left(\frac{\mu_{sim}}{\mu_{obs}} - 1\right)^2} \,, \qquad (3)$$

where $\sigma_{obs}$ is the standard deviation in observations, $\sigma_{sim}$ the standard deviation in simulations, $\mu_{sim}$ the simulation mean, and 10 $\mu_{obs}$ the observation mean (i.e. equivalent to $\overline{Q_{obs}}$). Like NSE, KGE = 1 indicates perfect agreement between simulations and observations. Analogous to NSE = 0, certain authors state that KGE < 0 indicates that the mean of observations provides better estimates than simulations (Castaneda-Gonzalez et al., 2018; Koskinen et al., 2017), although others state that this interpretation should not be attached to KGE = 0 (Gelati et al., 2018; Mosier et al., 2016). Various authors use positive KGE values as indicative of 'good' model simulations, whereas negative KGE values are considered 'bad', without explicitly 15 indicating that they treat KGE = 0 as their threshold between 'good' and 'bad' performance. For example, Rogelis et al (2016) consider model performance to be 'poor' for 0.5 > KGE > 0, and negative KGE values are not mentioned. Schönfelder et al (2017) consider negative KGE values 'not satisfactory'. Andersson et al (2017) mention negative KGE values in the same sentence as negative NSE values, implying that both are considered similarly unwanted. Fowler et al (2018) consider reducing the number of occurrences of negative KGE values as desirable. Knoben et al. (2018) cap figure legends at KGE = 0 and mask 20 negative KGE values. Siqueira et al (2018) consider ensemble behaviour undesirable as long as it produces negative KGE and NSE values. Sutanudjaja et al (2018) only count catchments where their model achieves KGE > 0 as places where their model application was successful. Finally, Towner et al (2019) use KGE = 0 as the threshold to switch from red to blue colour coding of model results, and only positive KGE values are considered 'skilful'. Naturally, authors prefer higher efficiency values over lower values, because this indicates their model is closer to perfectly reproducing observations (i.e. KGE = 1). Considering 25 the traditional use of NSE and its inherent quality that the mean flow results in NSE = 0, placing the threshold for 'good' model performance at KGE = 0 seems equally natural. We show in this paper that this reasoning is generally correct – positive KGE values do indicate improvements upon the mean flow benchmark – but not complete. In KGE terms, negative values do not necessarily indicate a model that performs worse than the mean flow benchmark. We first show this in mathematical terms and then present results from a synthetic experiment to highlight that NSE and KGE values are not directly comparable and 30 that understanding of the NSE metric does not translate well into understanding of the KGE metric.

Note that a weighted KGE version exists that allows specification of the relative importance of the three KGE terms (Gupta et al., 2009), as do a modified KGE (Kling et al., 2012) and a non-parametric KGE (Pool et al., 2018). These are not explicitly discussed here, because the issue we address here (i.e. the lack of an inherent benchmark in the KGE equation) applies to all these variants of KGE.

35 **2 KGE value of the mean flow benchmark**

Consider the case where $Q_{sim}(t) = \overline{Q_{obs}}$ for an arbitrary number of time steps, and where $\overline{Q_{obs}}$ is calculated from an arbitrary observed hydrograph. In this particular case, $\mu_{obs} = \mu_{sim}$, $\sigma_{obs} \neq 0$ but $\sigma_{sim} = 0$. Although the linear correlation between observations and simulations is formally undefined when $\sigma_{sim} = 0$, it makes intuitive sense to assign $r = 0$ in this case, since there is no relationship between the fluctuations of the observed and simulated hydrographs. Equation (3) becomes (positive 40 terms shown as symbols):

$$KGE = 1 - \sqrt{(0-1)^2 + \left(\frac{0}{\sigma_{obs}} - 1\right)^2 + \left(\frac{\mu_{obs}}{\mu_{obs}} - 1\right)^2} \, , \tag{4}$$

$$KGE = 1 - \sqrt{(0-1)^2 + (0-1)^2 + (1-1)^2} \, , \tag{5}$$

$$KGE = 1 - \sqrt{2} \, , \tag{6}$$

Thus, the KGE score for a mean flow benchmark is $KGE(\overline{Q_{obs}}) \approx -0.41$.

**3 Consequences**

**3.1 NSE and KGE values cannot be directly compared and should not be treated as approximately equivalent**

Through long use, hydrologic modellers have developed intuitive assessments about which NSE values can be considered acceptable for their preferred model(s) and/or catchment(s); however, this interpretation of acceptable NSE values cannot easily be mapped onto corresponding KGE values. There is no unique relationship between NSE and KGE values (Figure 1a,

note the scatter along both axes, see also Appendix 1) and where NSE values fall in the KGE component space depends in part on the coefficient of variation (CV) of the observations (see animated Figure S1 in Electronic Supplement 1 for a comparison of where NSE = 0 and $KGE = 1 - \sqrt{2}$ fall in the space described by KGE's r, a and b components for different CVs, highlighting that many different combinations of r, a and b can result in the same overall NSE or KGE value).

This has important implications when NSE or KGE thresholds are used to distinguish between behavioural and non-

15 behavioural models (that is, when a threshold is used to decide between accepting or rejecting models). Figure 1b-g are used to illustrate a synthetic experiment, where simulated flows are generated from observations and a threshold for behavioural models is set midway between the value for the mean flow benchmark (NSE=0 and KGE=-0.41) and the value for a perfect simulation (NSE=KGE=1): simulations are considered behavioural if NSE > 0.5 or KGE > 0.3. Each row shows flows from a different catchment, with increasing coefficients of variations (i.e. 0.28, 2.06 and 5.00 respectively). In Figures 1b, 1d and 1f,

the simulated flow is calculated as the mean of observations. NSE values are constant at NSE = 0 for all three catchments, and KGE values are constant at KGE = -0.41. In Figures 1c, 1e and 1g, the simulated flow is the observed flow plus an offset, to demonstrate the variety of impacts that bias has on NSE and KGE (similar examples could be generated for other types of error relating to correlation or variability, but these examples are sufficient to make the point that NSE and KGE behave quite differently). In Figure 1c, simulated flows are calculated as observed flows +0.45 mm/d (bias +39%). With the specified

thresholds, this simulation would be considered behavioural when using KGE (0.61 > 0.3), but not with NSE (-0.95 < 0.5). In Figure 1e, simulated flows are calculated as observed flows +0.5 mm/d (bias +40%). In this case, however, these simulations are considered behavioural with both metrics (NSE: 0.96 > 0.5; KGE: 0.60 > 0.3). Figure 1g shows an example where simulated flows are calculated as observations +0.7 mm/d (bias +97%), which is considered behavioural when NSE is used (0.96 > 0.5), but not when KGE is used (0.03 < 0.3).

These figures show that NSE values that are traditionally interpreted as high do not necessarily translate into high KGE values, and that standards of acceptability developed through extensive use of the NSE metric are not directly applicable to KGE values. Instead, hydrologists who choose to use the KGE metric need to develop new understanding of how this metric should be interpreted and not let themselves be guided by their understanding of NSE.

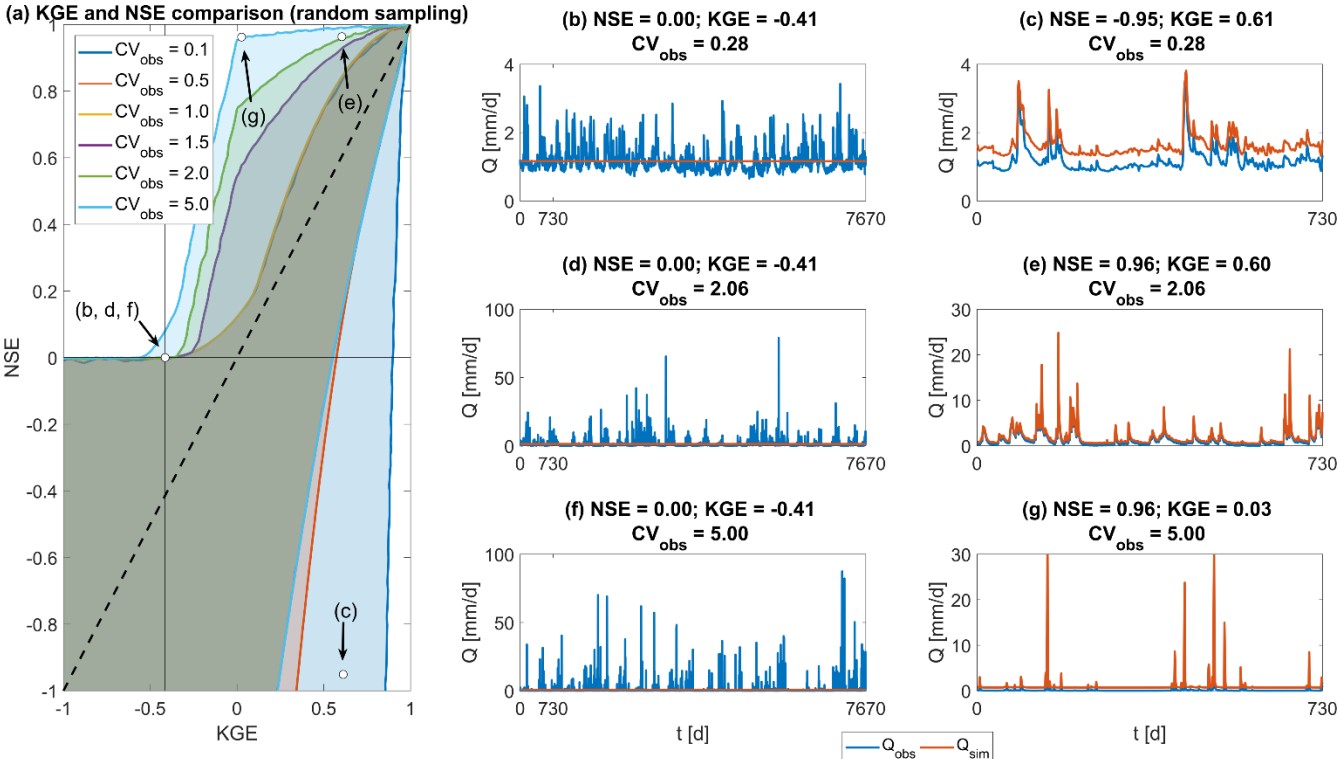

**Figure 1: Overview of the relationship between NSE and KGE. (a) Comparison of KGE and NSE values based on random sampling of the r, a and b components used in KGE and NSE, using 6 different values for the Coefficient of Variation of observations (see appendix for method and separate plots of each plane). Internal axes are drawn at KGE = 1-√2 and NSE = 0. The dashed diagonal is the 1:1 line. Locations of figures b-g indicated in brackets. (b, d, f) Simulated flow Q$_{sim}$ is created from the mean of Q$_{obs}$. (c) Q$_{sim}$ is created as Q$_{obs}$+0.45 mm/d on every time step, increasing the bias of observations. (e) Q$_{sim}$ is created as Q$_{obs}$+0.5 mm/d on every time step. (g) Q$_{sim}$ is created as Q$_{obs}$+0.7 mm/d on every time step. The y-axis is capped at 30 mm/d to better visualise the difference between observations and synthetic simulations.**

### 3.2 Explicit statements about benchmark performance are needed in modelling studies

The Nash-Sutcliffe Efficiency has an inherent benchmark in the form of the mean flow, giving NSE = 0. This benchmark is not inherent in the definition of the Kling-Gupta Efficiency, which is instead an expression of distance away from the point of ideal model performance in the space described by its three components. When $Q_{sim}$ is $\overline{Q_{obs}}$, $KGE \approx -0.41$, but there is no direct reason to choose this benchmark over other options (see e.g. Ding, 2019; Schaefli and Gupta, 2007; Seibert, 2001; Seibert et al., 2018). Because KGE itself has no inherent benchmark value to enable a distinction between 'good' and 'bad' models, modellers using KGE must be explicit about the benchmark model or value they use to compare the performance of their model against. As succinctly stated in Schaefli and Gupta (2007): *"Every modelling study should explain and justify the choice of benchmark [that] should fulfil the basic requirement that every hydrologist can immediately understand its explanatory power for the given case study and, therefore, appreciate how much better the actual hydrologic model is."*

If the mean flow is chosen as a benchmark, model performance in the range -0.41 < KGE <= 1 could be considered 'reasonable' in the sense that the model outperforms this benchmark. By artificially and consistently imposing a threshold at KGE = 0 to distinguish between 'good' and 'bad' models, modellers limit themselves in the models and/or parameter sets they consider in a given study, without rational justification of this choice and without taking into account whether more catchment-appropriate or study-appropriate thresholds could be defined.

### 3.3 On communicating model performance through skill scores

If the benchmark is explicitly chosen then a so-called skill score can be defined, which is the performance of any model compared to the pre-defined benchmark (e.g. Hirpa et al., 2018; Towner et al., 2019):

$$KGE_{skill\ score} = \frac{KGE_{model} - KGE_{benchmark}}{1 - KGE_{benchmark}}$$

The skill score is scaled such that positive values indicate a model that is better than the benchmark model and negative values indicate a model that is worse than the benchmark model. This has a clear benefit in communicating whether a model improves on a given benchmark or not with an intuitive threshold at $KGE_{skill\ score} = 0$, where negative values clearly indicate a model

worse than the benchmark and positive values a model that outperforms the benchmark.

However, scaling the KGE metric might introduce a different communication issue. In absolute terms, it seems clear that improving on $KGE_{benchmark} = 0.99$ by using a model might be difficult: the "potential for model improvement over benchmark" is only $1-0.99 = 0.01$. With a scaled metric, the "potential for model improvement over benchmark" always has range [0,1] but information about how large this potential was in the first place is lost and must be reported separately for proper context. If

the benchmark is already very close to perfect simulation, a $KGE_{skill\ score}$ of 0.5 might indicate no real improvement in practical terms. In cases where the benchmark constitutes a poor simulation, a $KGE_{skill\ score}$ of 0.5 might indicate a large improvement through using the model. This issue applies to any metric that is converted to a skill score.

Similarly, a skill score reduces the ease of communication about model deficiencies. It is generally difficult to interpret any score above the benchmark score but below the perfect simulation (in case of the KGE metric, $KGE = 1$) beyond 'higher is

better', but an absolute KGE score can at least be interpreted in terms of deviation-from-perfect on its a, b and r components. A score of $KGE = 0.95$ with $r = 1$, $a = 1$ and $b = 1.05$ indicates simulations with 5% bias. The scaled $KGE_{skill\ score} = 0.95$ cannot so readily be interpreted.

### 3.4 The way forward: new understanding based on purpose-dependent metrics and benchmarks

The modelling community currently does not have a single perfect model performance metric that is suitable for every study

purpose. Indeed, global metrics that attempt to lump complex model behaviour and residual errors into a single value may not be useful for exploring model deficiencies and diagnostics into how models fail or lack certain processes. If such metrics are used however, a modeller should make a conscious and well-founded choice about which aspects of the simulation they consider most important (if any), and in which aspects of the simulation they are willing to accept larger errors. The model's performance score should then be compared against an appropriate benchmark, which can inform to what extent the model is

fit for purpose.

If the KGE metric is used, emphasizing certain aspects of a simulation is straightforward by attaching weights to the individual KGE components to reduce or increase the impact of certain errors on the overall KGE score, treating the calibration as a multi-objective problem (e.g. Gupta et al., 1998) with varying weights assigned to the three objectives. An example of the necessity of such an approach can be found in Fig. 1g. For a study focussing on flood peaks, an error of only 0.7 mm/d for

each peak might be considered skilful, although the bias of these simulations is very large (+97%). Due to the small errors and the high coefficient of variation in this catchment, the NSE score of these simulations reaches a value that would traditionally be considered as very high (NSE = 0.96). The standard formulation of KGE however is heavily impacted by the large bias and the simulations in Fig. 1g result in a relatively low KGE score (KGE = 0.03). If one relies on this aggregated KGE value only, the low KGE score might lead a modeller to disqualify these simulations from further analysis, even if the simulations are

performing very well for the purpose of peak flow simulation. Investigation of the individual components of KGE would show that this low value is only due to bias errors and not due to an inability to simulate peak flows. The possibility to attach different weights to specific components of the KGE metric can allow a modeller to shift the metric's focus: by reducing the importance of bias in determining the overall KGE score, or emphasizing the importance of the flow variability error, the metric's focus can be moved towards peak flow accuracy (see Mizukami et al., 2019 for a discussion of purpose-dependent KGE weights and

a comparison between (weighted) KGE and NSE for high-flow simulation). For example, using weightings [1,5,1] for [r,a,b] to emphasize peak flow simulation (following Mizukami et al., 2019), the KGE score in Fig. 1g would increase to KGE = 0.81

This purpose-dependent score should then be compared against a purpose-dependent benchmark to determine whether the model can be considered fit for purpose.

However, aggregated performance metrics with a statistical nature, such as KGE, are not necessarily informative about model deficiencies from a hydrologic point of view (Gupta et al., 2008). While KGE improves upon the NSE metric in certain ways, Gupta et al. (2009) explicitly state that their intent with KGE was *"not to design an improved measure of model performance"* but only to use the metric to illustrate that there are inherent problems with mean-squared-error-based optimization approaches. They highlight an obvious weakness of the KGE metric, namely that many hydrologically relevant aspects of model performance (such as the shape of rising limbs and recessions, as well as timing of peak flows) are all lumped into the single correlation component. Future work could investigate alternative metrics that separate the correlation component of KGE into multiple, hydrologically meaningful, aspects. There is no reason to limit such a metric to only three components either and alternative metrics (or sets of metric components) can be used to expand the multi-objective optimization from three components to as many dimensions as are considered necessary or hydrologically informative. Similarly, there is no reason to use aggregated metrics only and investigating model behaviour on the individual time-step level can provide increased insight in where models fail (e.g. Beven et al., 2014).

Regardless whether KGE or some other metric is used, the final step in any modelling exercise would be comparing the obtained efficiency score against a certain benchmark that dictates which kind of model performance might be expected (e.g. Seibert et al., 2018) and decide whether the model is truly skilful. These benchmarks should not be specified in an ad-hoc manner (e.g. our earlier example where the thresholds are arbitrarily set at NSE = 0.5 and KGE = 0.3 is decidedly poor practice) but should be based on hydrologically meaningful considerations. The explanatory power of the model should be obvious from the comparison of benchmark and model performance values (Schaefli and Gupta, 2007), such that the modeller can make an informed choice on whether to accept or reject the model, and make an assessment of the model's strengths and where current model deficiencies are present. Defining such benchmarks is not straightforward because it relies on the interplay between our current hydrologic understanding, the availability and quality of observations, the choice of model structure and parameter values, and modelling objectives. However, explicitly defining such well-informed benchmarks will allow more robust assessments of model performance (see for example Abramowitz, 2012, for a discussion of this process in the land-surface community). How to define a similar framework within hydrology is an open question to the hydrologic community.

## 4 Conclusions

There is a tendency in current literature to interpret Kling-Gupta Efficiency (KGE) values in the same way as Nash-Sutcliffe Efficiency (NSE) values: negative values indicate 'bad' model performance, whereas positive values indicate 'good' model performance. We show that the traditional mean flow benchmark that results in NSE = 0 and the likely origin of this 'bad/good' model distinction, results in $KGE = 1 - \sqrt{2}$. Unlike NSE, KGE does not have an inherent benchmark against which flows are compared and there is no specific meaning attached to KGE = 0. Modellers using KGE must be specific about the benchmark against which they compare their model performance. If the mean flow is used as a KGE benchmark, all model simulations with -0.41 < KGE ≤ 1 exceeds this benchmark. Furthermore, modellers must take care to not let their interpretation of KGE values be consciously or subconsciously guided by their understanding of NSE values, because these two metrics cannot be compared in a straightforward manner. Instead of relying on the overall KGE value, in-depth analysis of the KGE components can allow a modeller to both better understand what the overall value means in terms of model errors and to modify the metric through weighting of the components to better align with the study's purpose. More generally, a strong case can be made for moving away from ad-hoc use of aggregated efficiency metrics and towards a framework based on purpose-dependent evaluation metrics and benchmarks that allows for more robust model adequacy assessment.

**Appendix 1**

The relation between possible KGE and NSE values shown in Figure 1a have been determined through random sampling of 1000000 different combinations of the components r, a and b of KGE (Eq. 2), for 6 different coefficients of variation (CV; 0.1, 0.5, 1.0, 1.5, 2.0, 5.0 respectively). Values were sampled in the following ranges: r = [-1,1]; a = [0,2]; b = [0,2]. The KGE value of each sample is found through Equation 2. The corresponding NSE value for each sampled combination of r, a and b is found through:

$$NSE = 2ar - a^2 - \frac{(b-1)^2}{CV_{obs}^2}, \tag{7}$$

Figure 2 shows the correspondence between KGE and NSE values for the 6 different CVs. Axis limits have been capped at [-1,1] for clarity. Equation 7 can be found by starting from Equation 4 in Gupta et al (2009) and expressing $\beta_n = \frac{\mu_s - \mu_o}{\sigma_o}$ in terms of $b = \frac{\mu_s}{\mu_o}$, using $CV_{obs} = \frac{\sigma_{obs}}{\mu_{obs}}$.

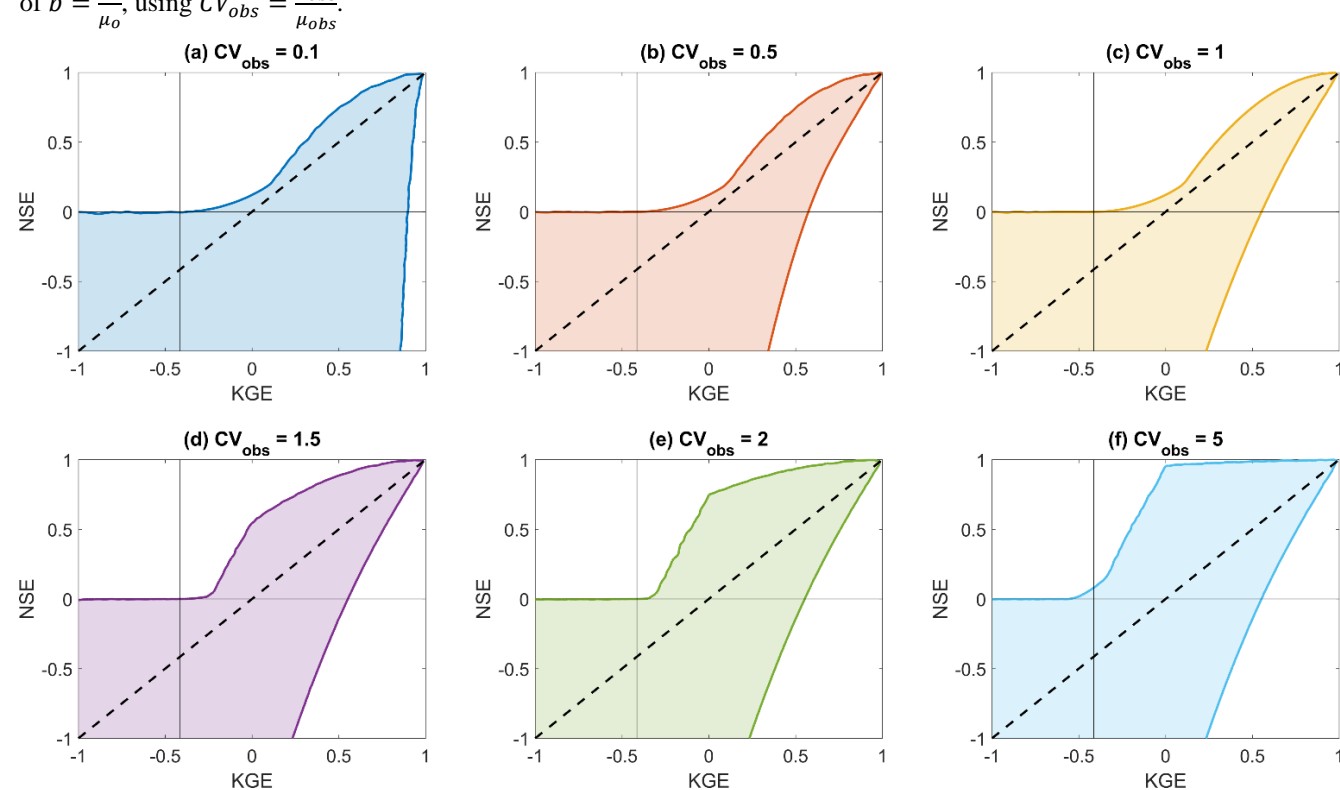

**Figure 2: Correspondence between synthetic KGE and NSE values based on 1E6 random samples of components r, a and b, for different coefficients of variation (CV). Colour coding corresponds to the colours used in Figure 1a.**

**Acknowledgements**

This work was funded by the Engineering and Physical Sciences Research Council WISE CDT, grant reference number EP/L016214/1. We are grateful to Hoshin Gupta, John Ding, Paul Whitfield and one anonymous reviewer, for their time and comments which helped us strengthen the message presented in this work.

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
