# Peer review of "Technical note: Inherent benchmark or not? Comparing Nash-Sutcliffe and Kling-Gupta efficiency scores"

_Hydrology and Earth System Sciences, 2019_

## Short Comment (SC1) · 11 Jul 2019

Equating the NSE and KGE scores

The authors raise an interesting question of whether or not the mean observed flow is an inherent benchmark of the NSE and KGE criteria.

The mean flow is a base value intended by Nash and Sutcliffe (1970) to scale their NSE score to between 0 and 1. Corresponding KGE scores are -0.41 and 1 (Page 3,

Line 10). Rescaling the KGE criterion to (KGE+0.41)/1.41 would produce a 0 to 1 scale.

While worth searching for "a single perfect (hydrologic) model performance metric" (Page 4, Line 10), equally important if not more, in my opinion, is finding an alternate "starter" model to the mean flow one, the "no model" one in NSE. This will be a new benchmark or baseline against which the performances of other hydrologic models are to be measured.

One of the "least skill(ful)" ones is a one–step linear extrapolation model of the observed flows. The predicted or forecast flow by extrapolation is: $Q_{fore}(t) = Q_{obs}(t-1) + [Q_{obs}(t-1) - Q_{obs}(t-2)]$. This is a simplest autoregressive model of order 2. It has been used on its own, i.e., outside the NSE, as a river forecast model.

The NSE criterion may be modified by substituting the mean observed flow term, $\overline{Q_{obs}}$, in Equation (1), by the forecast flow. See Mizukami et al. (2019) cited by the authors for my previous comment on this (SC1 therein), the deficiency of the extrapolation model included.

---

## Referee Comment (RC1) · Hoshin Gupta (Referee) · 23 Jul 2019

Attached as a pdf file.

Please also note the supplement to this comment:
https://www.hydrol-earth-syst-sci-discuss.net/hess-2019-327/hess-2019-327-RC1-supplement.pdf
* * *
[Figure]

**Review of HESS Technical note: "*Inherent benchmark or not? Comparing Nash- Sutcliffe and Kling-Gupta efficiency scores*", by Wouter J, M Knoben, JE Freer and RA Woods**

**Review Provided by Hoshin Gupta (23rd July 2019)**

**Summary of the Paper:** The paper makes perhaps three main points:

**Main Point Number (1): On Use of the "Mean Flow Benchmark" to interpret NSE and KGE**

- The NSE normalizes model performance to an interpretable scale such that NSE = 1 indicates perfect correspondence between simulations and observations, NSE = 0 indicates that the model simulations have the same explanatory power as the mean of the observations, and NSE < 0 indicates that the model is a worse predictor than the mean of the observations.

- NSE = 0 is regularly used as a benchmark to distinguish '*good*' and '*bad*' models, although this threshold could be considered a low level of predictive skill and is also a relatively arbitrary choice.

- KGE addresses several shortcomings in NSE and is increasingly used for model calibration and evaluation. Like NSE, KGE = 1 indicates perfect agreement between simulations and observations.

- Some users have tried to assign a similar scale/threshold as with NSE to be used in interpretation of KGE scores. Many authors use positive KGE values as indicative of '*good*' model performance, and negative KGE values as indicative of '*bad*' performance.

- However, this paper shows that placing the threshold for '*good*' model performance at KGE = 0 is generally correct (i.e., positive KGE values do indicate improvements upon the mean flow benchmark) but not complete. In fact, *negative KGE values do not necessarily indicate a model that performs worse than the mean flow benchmark*. The authors show this in mathematical terms, and then present results from a synthetic experiment to highlight that NSE and KGE values are not directly comparable and that understanding of the NSE metric does not translate well into understanding of the KGE metric.

- Mathematically, if the model simulations of the system responses are in fact constant over time and equal to the mean of the observed flows (the mean flow benchmark), we actually have KGE ≈ −0.41.

**Main Point Number (2): On the Need to Explicitly Consider Benchmark Performance**

- NSE and KGE values cannot be directly compared and should not be treated as approximately equivalent. There is no unique relationship between NSE and KGE values and where NSE values fall in the KGE component space depends in part on the coefficient of variation (CV) of the observations.

- NSE values that are traditionally seen as high do not necessarily translate into high KGE values. Hydrologists who choose to use the KGE metric need to develop new understanding of how this metric should be interpreted and not let themselves be guided by their understanding of NSE.

**Fig. 1.**

**Supplement:**

**Review of HESS Technical note: "*Inherent benchmark or not? Comparing Nash- Sutcliffe and Kling-Gupta efficiency scores*", by Wouter J, M Knoben, JE Freer and RA Woods**

**Review Provided by Hoshin Gupta (23rd July 2019)**

**Summary of the Paper:** The paper makes perhaps three main points:

**Main Point Number (1): On Use of the "Mean Flow Benchmark" to interpret NSE and KGE**

- The NSE normalizes model performance to an interpretable scale such that NSE = 1 indicates perfect correspondence between simulations and observations, NSE = 0 indicates that the model simulations have the same explanatory power as the mean of the observations, and NSE < 0 indicates that the model is a worse predictor than the mean of the observations.

- NSE = 0 is regularly used as a benchmark to distinguish '*good*' and '*bad*' models, although this threshold could be considered a low level of predictive skill and is also a relatively arbitrary choice.

- KGE addresses several shortcomings in NSE and is increasingly used for model calibration and evaluation. Like NSE, KGE = 1 indicates perfect agreement between simulations and observations.

- Some users have tried to assign a similar scale/threshold as with NSE to be used in interpretation of KGE scores. Many authors use positive KGE values as indicative of '*good*' model performance, and negative KGE values as indicative of '*bad*' performance.

- However, this paper shows that placing the threshold for '*good*' model performance at KGE = 0 is generally correct (i.e., positive KGE values do indicate improvements upon the mean flow benchmark) but not complete. In fact, negative KGE values do not necessarily indicate a model that performs worse than the mean flow benchmark. The authors show this in mathematical terms, and then present results from a synthetic experiment to highlight that NSE and KGE values are not directly comparable and that understanding of the NSE metric does not translate well into understanding of the KGE metric.

- Mathematically, if the model simulations of the system responses are in fact constant over time and equal to the mean of the observed flows (the mean flow benchmark), we actually have KGE ≈ −0.41.

**Main Point Number (2): On the Need to Explicitly Consider Benchmark Performance**

- NSE and KGE values cannot be directly compared and should not be treated as approximately equivalent. There is no unique relationship between NSE and KGE values and where NSE values fall in the KGE component space depends in part on the coefficient of variation (CV) of the observations.

- NSE values that are traditionally seen as high do not necessarily translate into high KGE values. Hydrologists who choose to use the KGE metric need to develop new understanding of how this metric should be interpreted and not let themselves be guided by their understanding of NSE.

- Whereas NSE has an inherent benchmark in the form of the mean flow, this benchmark is not inherent in the definition of KGE, which is instead an expression of distance away from the point of ideal model performance in the space described by its three components.

- There is no direct reason to use the "*mean flow*" as a benchmark over other options.

- Because KGE has no inherent benchmark value to enable a distinction between '*good*' and '*bad*' models, *modelers using KGE must be explicit about the benchmark model or value they use to compare the performance of their model against*.

- By choosing the mean flow as a benchmark to distinguish between '*good*' and '*bad*' models, practitioners limit themselves in the models and/or parameter sets they consider in a given study, without rational justification.

**Main Point Number (3):  On the Need to Recognize that Metrics and Benchmarks are Purpose-Dependent**

- There is no single perfect model performance metric that is suitable for every study purpose. Indeed, global metrics that lump complex model behaviour and residual errors into a single value are not useful for exploring model deficiencies and diagnostics regarding how models fail or lack certain processes.

- In the choice of metrics, modellers should make conscious and well-founded choices about which aspects of the simulation they consider most important (if any), and in which aspects of the simulation they are willing to accept larger errors.

- When using KGE, emphasizing certain aspects of a simulation is straightforward by attaching weights to the individual KGE components to reduce or increase the impact of certain errors on the overall KGE score.

- This purpose-dependent score should then be compared against a purpose-dependent benchmark to determine whether the model can be considered '*good*'.

- How these purpose-dependent benchmarks should be set is an open question to the hydrologic community.

**My Review Remarks:**

[1] I thoroughly enjoyed reading this Technical Note contribution by *Wouter, Knoben, Freer* and *Woods*, and I thank them for (re)raising some very important issues, and for their new/original contribution regarding the value that the KGE criterion takes on when using the mean flow as a benchmark.  As such, I have no critique per se to offer regarding this paper, and compliment the authors on an excellent contribution to the literature.

[2] Instead I would like to focus on some interesting points raised by this work.  This review opportunity allows me to take the liberty of reminding the readers of some interesting points that were previously raised in *Schaefli and Gupta (2007)* and *Gupta et al (2009)*, that the authors

allude to, _but which perhaps could be strengthened by the authors of the current work in their presentation_.  Text between quotes is reproduced from the original papers.

[3] Beginning first with _Schaefli and Gupta (2007)_, that paper was about benchmarking.  In it, we discussed the fact that the process by which anyone assesses and communicates model performance evaluation is of primary importance, and that "_the basic 'rule' is that every modelling result should be put into context, for example, by indicating the model performance using appropriate indicators, and by highlighting potential sources of uncertainty_".

[4] We pointed out therein (as have others before and after us) that:

   a) the "_NSE value, while a convenient and normalized measure of model performance does not provide a reliable basis for comparing the results of different case studies_"
   b) the "_use of the mean observed value as a reference can be a very poor predictor (e.g. for strongly seasonal time series), or a relatively good predictor (e.g. for time series that are essentially fluctuations around a relatively constant mean value)_".

For example, "_In the case of strongly seasonal time series, a model that only explains the seasonality but fails to reproduce any smaller time scale fluctuations will report a good NSE value; for predictions at the daily time step, this (high) value will be misleading. In contrast, if the model is intended to simulate the fluctuations around a relatively constant mean value, it can only achieve high NSE values if it explains the small time-scale fluctuations_".

[5] Therefore, the definition of an appropriate benchmark model is particularly important … to properly communicate how good a model really is, it is necessary to establish an appropriate reference (or benchmark model) for a given case study and a given modelling time step. In that paper we mention some examples, including:

   a) the interannual mean value for every calendar day proposed by _Garrick et al. (1978)_ for systems having strong but relatively constant seasonality
   b) a simple adjusted precipitation benchmark (APB) where the rainfall is scaled to match the mean discharge and shifted in time by some optimum lag that reflects the time of concentration of the basin, and
   c) a smoothed version of the APB where a simple dispersion process (moving average) is added to adjust the smoothness of the scaled-down and translated precipitation to match the smoothness of the observed discharge, for example by maximizing the correlation between the adjusted precipitation and the observed flow _(Morin et al., 2002)_.

Of course, many other possible benchmarks can be conceived, such as "_persistence_" (the next time steps' simulated flow is the same as the current time step's observed flow), some kind of linear or non-linear extrapolation into the future, and some kind of data-based time-series analytical model projection such as can be constructed by ARMAX or ANN methods.

[6] In the conclusions to _Schaefli and Gupta (2007)_, we argued that the definition of an appropriate baseline for model performance, and in particular, for measures such as NSE (and by extension, KGE or any other model performance measure), should become part of the 'best practices' in hydrologic modelling, that "_Every modelling study should explain and justify the choice of benchmark_", and that "_the benchmark should fulfill the basic requirement that every_

*hydrologist can immediately understand its explanatory power for the given case study and, therefore, appreciate how much better the actual hydrologic model is*".

[7] Moving next to *Gupta et al (2009)*, we discussed the fact that the NSE, which is a dimensionless mathematical normalization of the mean squared error (MSE) criterion can be viewed as a classic skill score (*Murphy, 1988*), where 'skill' is interpreted as the comparative ability of a model with regards to a baseline 'model'. Further, as shown by *Murphy (1988)* and *Weglarczyk (1998)*, it is possible to decompose the NSE criterion into components (correlation, conditional bias, and unconditional bias) that facilitates a better understanding of what is causing a particular model performance to be 'good' or 'bad', while providing insight into possible trade-offs between the different components.

[8] Our own particular diagnostic decomposition of NSE (and hence MSE) was developed in the context of our interest in hydrological modelling where, as we showed, interactions among these components (correlation, mean bias, and variance bias) can cause problems during model calibration – possibly leading to parameter estimates that are associated with large volume balance errors and/or underestimation of the variability in the flows. Further, we pointed out that many different combinations of the three components can result in the same overall value for NSE, leading to considerable ambiguity in the comparative evaluation of alternative model hypotheses.

[9] Importantly, we also pointed out that, rather than trying to come up with a 'corrected' version of the NSE criterion, *the whole calibration problem can instead be viewed from the multi-objective perspective* (see e.g., *Gupta et al., 1998*), by focusing on the correlation, variability error and bias error as separate criteria to be optimized. *When we do so, if a compromise solution is desired, we can use the solution provided by the KGE or one of its alternatively weighted variants*.

[10] We presented some comparative experimental results that show that when optimizing on KGE, there is a strong correlation between the values obtained for the KGE and NSE criteria, but when optimizing on NSE, the correlation between the values obtained for NSE and KGE is lower due to the fact that optimization on KGE strongly controls the values that the mean and variance ratio components can achieve, whereas optimization on NSE constrains these components only weakly. Overall, the use of KGE instead of NSE for model calibration tends to improve the bias and variability measures considerably while only slightly decreasing the correlation.

[11] Finally, we pointed out that the NSE/MSE or KGE performance metric decomposition relates to the idea of diagnostic model evaluation, as proposed by *Gupta et al. (2008)*, *which is to move beyond aggregate measures of model performance that are primarily statistical in meaning, towards the use of (multiple) measures and signature plots that are selected for their ability to provide hydrological interpretation*. While the theoretical development behind the KGE provides one simple, statistically founded approach to the development of a strategy for diagnostic evaluation and calibration of a model, *we also pointed out that all other statistical properties beyond the mean and standard deviation (which are two long-term statistics of the data), such as timing of the peaks, and shapes of the rising limbs and the recessions of the hydrograph (i.e. autocorrelation structures), are lumped into the (linear) correlation coefficient as an aggregate measure*.

[12] We therefore suggested that a logical next step would be to consider other relevant diagnostic properties (such as for example, different aspects of flow timing and shape), but left those considerations are left for future work. For example, although not mentioned explicitly in *Gupta et al (2009)*, there is no reason that other (statistical or otherwise) aspects of model performance, such as "*skewness*", or "*particular quantiles*" etc., should not be integrated into the basis for model performance evaluation and, if desired, built into a "*KGE-like*" metric.

[13] However, the _explicitly stated purpose_ of the *Gupta et al (2009)* study _was NOT to design an improved measure of model performance_, but instead:

a) to show clearly that there are systematic problems inherent with any optimization that is based on mean squared errors (such as NSE),

b) that "*the alternative criterion KGE was simply used for illustration purposes*" (many different alternative criteria would also be sensible), and

c) that "*Ultimately the decision to accept or reject a model must be made by an expert hydrologist, where such a decision is best based in a multiple-criteria framework*", where tracking the mean bias, variance bias and correlation (and other possible) components can help.

**Concluding Remarks:**

[14] With this context, it would actually be useful for the community to strategically move beyond the use of single metrics for model performance assessment (and/or selection), whether NSE or KGE or any other that might be conceived, and to follow the spirit of *Gupta et al (2008)* by designing some reasonable and rational basis for selecting "*sets*" of metrics that provide meaningful diagnostic evaluation of a model.

[15] As pointed out by the current authors, to be meaningful, any such metrics should be accompanied by meaningful benchmarks. To be meaningful, these benchmarks should _not_ be specified in an ad-hoc manner (such as NSE > 0.5 etc.) but should have some meaningful theoretical basis that conveys useful information to the decision maker.

[16] Indeed, I have often been contacted by researchers asking for some "*threshold*" values to use with KGE in their studies, and have always responded by discouraging such a practice and instead encouraging the use of the individual diagnostic components of KGE (and others that might be imagined) and setting associated thresholds using some meaningful basis.

[17] I do understand that, when performing studies involving large samples of data and/or many models, there is a tendency to want to use simple "*aggregate*" metrics in order to select or focus on a sub-set of "*good*" or "*poor*" models. However, there is arguably little to be gained by doing so by following the (arguably lazy) approach of using an aggregate metric that is not meaningfully interpretable.

[18] I sincerely hope that this current authored contribution will help to move the bulk of the community of hydrologic practitioners in the direction of using a more informative, and powerful, *diagnostic* (and necessarily multi-criteria) basis for model evaluation _that points to the nature of model deficiencies_ and therefore to the modeling issues that need fixing.

[19] It might be helpful therefore, for the current authors to make some stronger arguments/comments in this direction, to encourage movement beyond the use of NSE and/or KGE, and thereby to a more powerful and robust approach to model assessment, as has been (slowly) pursued the case in some closely related communities (*Abramowitz 2012*).

**References Cited:**

Abramowitz G, 2012, Towards a public, standardized, diagnostic benchmarking system for land surface models, Geoscientific Model Development, vol. 5, pp. 819 - 827, http://dx.doi.org/10.5194/gmd-5-819-2012

Garrick M, Cumane C, Nash JE. 1978. A criterion of efficiency for rainfall-runoff models. Journal of Hydrology 36: 375–381.

Gupta HV, S Sorooshian and PO Yapo (1998), Towards Improved Calibration of Hydrologic Models: Multiple and Non-Commensurable Measures of Information, Water Resources Research, Vol. 34, No. 4, pp. 751-763

Gupta HV, T Wagener and YQ Liu (2008), Reconciling Theory with Observations: Towards a Diagnostic Approach to Model Evaluation, Hydrological Processes, Vol. 22 (18), pp. 3802-3813, DOI: 10.1002/hyp.6989.

Gupta HV, H Kling, KK Yilmaz and GF Martinez-Baquero (2009), Decomposition of the Mean Squared Error & NSE Performance Criteria: Implications for Improving Hydrological Modelling, Journal of Hydrology, Vol. 377, pp. 80-91, doi: 10.1016/j.jhydrol.2009.08.003.

Morin E, Georgakakos KP, Shamir U, Garti R, Enzel Y. 2002. Objective, observations-based, automatic estimation of the catchment response timescale. Water Resources Research 38: 1212, DOI: 10·1029/2001WR000808.

Murphy A (1988), Skill scores based on the mean square error and their relationships to the correlation coefficient. Monthly Weather Review 116: 2417-2424

Schaefli B and HV Gupta (2007), Do Nash values have value? Hydrological Processes, 21(15), 2075-2080, simultaneously published online as Invited Commentary in Hydrologic Processes (HP Today), Wiley InterScience, doi: 10.1002/hyp.6825

Weglarczyk S (1998), The interdependence and applicability of some statistical quality measures for hydrological models. Journal of Hydrology 206: 98-103

---

## Referee Comment (RC2) · Anonymous Referee #2 · 4 Aug 2019

Summary:

The technical note provides interesting discussions on an interpretation of two metrics widely used in hydrologic community: NSE and KGE. First, the author reminds the readers that NSE is the metrics that quantify the performance compare to observed mean flow benchmark (NSE=0 indicates model performance is equivalent to this benchmark). The authors then state that there are many past studies that used KGE=0 as a threshold between bad and good model performance, same as NSE threshold. The authors point out KGE=0 does not hold the same meaning as NSE=0, and analytically show that KGE > -0.41 indicates that the model performs better than

observed mean flow (if a modeler compares the model to mean flow using KGE). The authors made a direct comparison between NSE and KGE by random sampling of each KGE component and corresponding NSE, showing there is no unique relationship between two metrics, but their range of NSE value given a KGE partly depends on Coefficient of variation of the observed flow, indicating NSE and KGE cannot be directly compared. Finally, the authors that single, aggregated metrics like NSE and KGE might be misleading if the modeler looks for a specific model application (i.e., flood forecast need accuracy of high flow), and the modelers need to look more targeted metrics related to the application.

Comment:

I agree on all the major statements made in this technical note. I think one Figure presented in the note is unique contribution. It is similar to Fig 6d Gupta et al.,2009, but is expanded version and generated in the different context. I think this is very informative article, and great particularly for hydrologic practitioners who tend to quickly and intuitively evaluate the model with either NSE or KGE. I did not find any corrections/suggestions I can offer and therefor I recommend publish as is.

---

## Author Comment (AC1) · 6 Aug 2019

Dear Dr John Ding,

Thank you for your comment on our manuscript and the reference to your earlier comments on this topic during the discussion of Mizukami et al (2019).

**Communication through scaled metrics**

We agree with your comment that KGE can be rescaled so that the KGE score of the mean flow equals 0. Both Feyera et al (2018) and Towner et al (2019) use a generalized scaled KGE as a skill score metric *[author's note: our thanks to Shaun Harrigan for pointing this out]*:

$$KGE_{skill\ score} = \frac{KGE_{model} - KGE_{benchmark}}{1 - KGE_{benchmark}}$$

This could potentially be of use for clearer communication of whether any model's KGE score exceeds the benchmark (i.e. all positive scores of $KGE_{skill\ score}$) or not (i.e. all negative scores on $KGE_{skill\ score}$).

However, scaling the KGE metric might introduce a different communication issue. In absolute terms, it seems clear that improving on $KGE_{benchmark}$ = 0.99 by using a model might be difficult: the "potential for model improvement over benchmark" is only 1-0.99 = 0.01. With a scaled metric, the "potential for model improvement over benchmark" always has range [0,1], but information about how large this potential was in the first place is lost and must be reported separately for proper context. If the benchmark is already very close to perfect simulation, a $KGE_{skill\ score}$ of 0.5 might indicate no real improvement in practical terms. In cases where the benchmark constitutes a poor simulation, a $KGE_{skill\ score}$ of 0.5 might indicate a large improvement through using the model.

Similarly, scaling the metric might also reduce the ease of communication about model deficiencies. It is generally difficult to interpret any score above the benchmark score but below the perfect simulation (1) beyond 'higher is better', but an absolute KGE score can at least be interpreted in terms of deviation-from-perfect on its a, b and r components (assuming they are also reported). A score of KGE = 0.95 with r = 1, a = 1 and b = 1.05 indicates simulations with 5% bias. A scaled KGE score of 0.95 cannot so readily be interpreted.

Therefore, we think that a scaled metric could be of use in some cases (the clear meaning of positive and negative values is useful) but also has some drawbacks: a scaled metric is not necessarily a more efficient way of communicating model performance (because still two values must be reported for proper context) and scaling also reduces the ease with which individual KGE components can be interpreted in terms of simulation deficiencies. We will consider adding these thoughts to the discussion section in our manuscript.

**Which benchmarks should be used?**

Communicating model performance in comparison to benchmark values is a separate issue from *which* benchmark should be used, which is the focus of the second part of your comment. We agree that the traditional mean flow benchmark is not a particularly taxing baseline in many (although not all) cases (as mentioned on page 1, lines 28-29) and that it is worthwhile to carefully consider alternative options (page 1, lines 28-30; page 3, lines 5-7). We already provide references to several other possible options for benchmarking (Schaefli and Gupta, 2007; Seibert, 2001; Seibert et al.,

2018) and will add your suggestion of a linear extrapolation model to this list. We intend to change the text to emphasize that (an) appropriate benchmark(s) should be chosen as part of the experimental design, such that model scores that outperform the benchmark are a clear reflection of the model being closer to the modelling aim than the benchmark was.

On behalf of all authors,

Kind regards,

Wouter Knoben

**References**

Feyera A. Hirpa, Peter Salamon, Hylke E. Beck, Valerio Lorini, Lorenzo Alfieri, Ervin Zsoter, Simon J. Dadson (2018). Calibration of the Global Flood Awareness System (GloFAS) using daily streamflow data. Journal of Hydrology, Volume 566, 595-606, https://doi.org/10.1016/j.jhydrol.2018.09.052

Mizukami, N., Rakovec, O., Newman, A. J., Clark, M. P., Wood, A. W., Gupta, H. V., & Kumar, R. (2019). On the choice of calibration metrics for "high-flow" estimation using hydrologic models. Hydrology and Earth System Sciences, 23(6), 2601–2614. https://doi.org/10.5194/hess-23-2601-2019

Schaefli, B. and Gupta, H. V. (2007). Do Nash values have value? Hydrol. Process., 21, 2075–2080, doi:10.1002/hyp.6825

Seibert, J. (2001). On the need for benchmarks in hydrological modelling. Hydrol. Process., 15(6), 1063–1064, doi:10.1002/hyp.446

Seibert, J., Vis, M. J. P., Lewis, E. and van Meerveld, H. J. (2018). Upper and lower benchmarks in hydrological modelling. Hydrol. Process., 32(8), 1120–1125, doi:10.1002/hyp.11476

Towner, J., Cloke, H. L., Zsoter, E., Flamig, Z., Hoch, J. M., Bazo, J., Coughlan de Perez, E., and Stephens, E. M. (in press). Assessing the performance of global hydrological models for capturing peak river flows in the Amazon Basin. Hydrol. Earth Syst. Sci. Discuss., https://doi.org/10.5194/hess-2019-44

Dear Dr John Ding,

Thank you for your comment on our manuscript and the reference to your earlier comments on this topic during the discussion of Mizukami et al (2019).

\textbf{Communication through scaled metrics}

We agree with your comment that KGE can be rescaled so that the KGE score of the mean flow equals 0. Both Feyera et al (2018) and Towner et al (2019) use a generalized scaled KGE as a skill score metric \emph{[author's note: our thanks to Shaun Harrigan for pointing this out]}:

\begin{equation}

KGE_{skill score} = \frac{KGE_{model}-KGE_{benchmark}}{1-KGE_{benchmark}}

\end{equation}

This could potentially be of use for clearer communication of whether any model's KGE score exceeds the benchmark (i.e. all positive scores of KGEskill score) or not (i.e. all negative scores on KGEskill score).

However, scaling the KGE metric might introduce a different communication issue. In absolute terms, it seems clear that improving on $KGE_{benchmark} = 0.99$ by using a model might be difficult: the "potential for model improvement over benchmark" is only 1-0.99 = 0.01. With a scaled metric, the "potential for model improvement over benchmark" always has range [0,1], but information about how large this potential was in the first place is lost and must be reported separately for proper context. If the benchmark is already very close to perfect simulation, a $KGE_{skill score}$ of 0.5 might indicate no real improvement in practical terms. In cases where the benchmark constitutes a poor simulation, a $KGE_{skill score}$ of 0.5 might indicate a large improvement through using the model.

Therefore, we think that a scaled metric could be of use in some cases (the clear meaning of positive and negative values is useful) but is not necessarily a more efficient way of communicating model performance (because still two values must be reported for proper context). We will consider adding these thoughts to the discussion section in our manuscript.

\textbf{Which benchmarks should be used?}

Communicating model performance in comparison to benchmark values is a separate issue from which benchmark should be used, which is the focus of the second part of your comment. We agree that the traditional mean flow benchmark is not a particularly taxing baseline in many (although not all) cases and (as mentioned on page 1, lines 28-29) and that it is worthwhile to carefully consider alternative options (page 1, lines 28-30; page 3, lines 5-7). We already provide references to several other possible options for benchmarking (Schaefli and Gupta, 2007; Seibert, 2001; Seibert et al., 2018) and will add your suggestion of a linear extrapolation model to this list.

On behalf of all authors,

Kind regards,

Wouter Knoben

\textbf{References}

Feyera A. Hirpa, Peter Salamon, Hylke E. Beck, Valerio Lorini, Lorenzo Alfieri, Ervin Zsoter, Simon J. Dadson (2018). Calibration of the Global Flood Awareness System (GloFAS) using daily streamflow data. Journal of Hydrology, Volume 566, 595-606, https://doi.org/10.1016/j.jhydrol.2018.09.052

Mizukami, N., Rakovec, O., Newman, A. J., Clark, M. P., Wood, A. W., Gupta, H. V., & Kumar, R. (2019). On the choice of calibration metrics for "high-flow" estimation using hydrologic models. Hydrology and Earth System Sciences, 23(6), 2601–2614. https://doi.org/10.5194/hess-23-2601-2019

Schaefli, B. and Gupta, H. V. (2007). Do Nash values have value? Hydrol. Process., 21, 2075–2080, doi:10.1002/hyp.6825

Seibert, J. (2001). On the need for benchmarks in hydrological modelling. Hydrol. Process., 15(6), 1063–1064, doi:10.1002/hyp.446

Seibert, J., Vis, M. J. P., Lewis, E. and van Meerveld, H. J. (2018). Upper and lower benchmarks in hydrological modelling. Hydrol. Process., 32(8), 1120–1125, doi:10.1002/hyp.11476

Towner, J., Cloke, H. L., Zsoter, E., Flamig, Z., Hoch, J. M., Bazo, J., Coughlan de Perez, E., and Stephens, E. M. (in press). Assessing the performance of global hydrological models for capturing peak river flows in the Amazon Basin. Hydrol. Earth Syst. Sci. Discuss., https://doi.org/10.5194/hess-2019-44

---

## Author Comment (AC2) · 7 Aug 2019

Dear reviewer,

Thank you for your kind words. We appreciate you taking the time to read this manuscript and providing us with this review.

Kind regards,

On behalf of all co-authors,

Wouter Knoben

---

## Author Comment (AC3) · 7 Aug 2019

Dear prof. Gupta,

Thank you for your kind words and this thought-provoking review. We will strengthen our comments in the direction you suggest. We intend to emphasize in our discussion that for model development and decision making, the community should consider moving beyond aggregated efficiency metrics and ad-hoc levels of acceptability and towards diagnostic metrics that show in which area(s) a model's deficiencies occur and towards metrics that can be interpreted from a hydrologic, rather than from a purely statistical point of view.

[Figure]

Kind regards,

On behalf of all co-authors,

Wouter Knoben

---

## Author Response (AR1)

Dear editor, dear reviewers,

Thank you for the time taken to review our contribution. We have revised our manuscript in response to your suggestions. This document shows our responses to your comments on a point-by-point basis. For clarity, our responses are given in blue. Page and line numbers of these changes are given in our responses in the format "P[page number]L[line number]" and refer to the track-changes manuscript.

In addition, we have corrected a few typographical errors in the manuscript and changed the order of our discussion paragraphs, so that they more logically flow from the problem at hand (NSE and KGE being treated as approximately equivalent) to potential ways to deal with this problem.

Contents:
- Response to review by H. Gupta
- Response to anonymous reviewer 2
- Response to comment by John Ding

Kind regards,

On behalf of all co-authors,

Wouter Knoben

[revised manuscript text omitted]

5 **Summary of the Paper:**

The paper makes perhaps three main points:

**Main Point Number (1): On Use of the "Mean Flow Benchmark" to interpret NSE and KGE**

• The NSE normalizes model performance to an interpretable scale such that NSE = 1 indicates perfect correspondence between simulations and observations, NSE = 0 indicates that the model simulations have the same

10 explanatory power as the mean of the observations, and NSE < 0 indicates that the model is a worse predictor than the mean of the observations.

• NSE = 0 is regularly used as a benchmark to distinguish '*good*' and '*bad*' models, although this threshold could be considered a low level of predictive skill and is also a relatively arbitrary choice.

• KGE addresses several shortcomings in NSE and is increasingly used for model calibration and evaluation. Like

15 NSE, KGE = 1 indicates perfect agreement between simulations and observations.

• Some users have tried to assign a similar scale/threshold as with NSE to be used in interpretation of KGE scores. Many authors use positive KGE values as indicative of '*good*' model performance, and negative KGE values as indicative of '*bad*' performance.

• However, this paper shows that placing the threshold for '*good*' model performance at KGE = 0 is generally

20 correct (i.e., positive KGE values do indicate improvements upon the mean flow benchmark) but not complete. In fact, *negative KGE values do not necessarily indicate a model that performs worse than the mean flow benchmark*. The authors show this in mathematical terms, and then present results from a synthetic experiment to highlight that NSE and KGE values are not directly comparable and that understanding of the NSE metric does not translate well into understanding of the KGE metric.

25 • Mathematically, if the model simulations of the system responses are in fact constant over time and equal to the mean of the observed flows (the mean flow benchmark), we actually have KGE $\approx -0.41$.

**Main Point Number (2): On the Need to Explicitly Consider Benchmark Performance**

• NSE and KGE values cannot be directly compared and should not be treated as approximately equivalent. There is no unique relationship between NSE and KGE values and where NSE values fall in the KGE component space

30 depends in part on the coefficient of variation (CV) of the observations.

• NSE values that are traditionally seen as high do not necessarily translate into high KGE values. Hydrologists who choose to use the KGE metric need to develop new understanding of how this metric should be interpreted and not let themselves be guided by their understanding of NSE.

• Whereas NSE has an inherent benchmark in the form of the mean flow, this benchmark is not inherent in the

35 definition of KGE, which is instead an expression of distance away from the point of ideal model performance in the space described by its three components.

• There is no direct reason to use the "*mean flow*" as a benchmark over other options.

• Because KGE has no inherent benchmark value to enable a distinction between '*good*' and '*bad*' models, *modelers using KGE must be explicit about the benchmark model or value they use to compare the performance of their model against*.

• By choosing the mean flow as a benchmark to distinguish between '*good*' and '*bad*' models, practitioners limit themselves in the models and/or parameter sets they consider in a given study, without rational justification.

**Main Point Number (3): On the Need to Recognize that Metrics and Benchmarks are Purpose- Dependent**

• There is no single perfect model performance metric that is suitable for every study purpose. Indeed, global metrics that lump complex model behaviour and residual errors into a single value are not useful for exploring model deficiencies and diagnostics regarding how models fail or lack certain processes.

• In the choice of metrics, modellers should make conscious and well-founded choices about which aspects of the simulation they consider most important (if any), and in which aspects of the simulation they are willing to accept larger errors.

• When using KGE, emphasizing certain aspects of a simulation is straightforward by attaching weights to the individual KGE components to reduce or increase the impact of certain errors on the overall KGE score.

• This purpose-dependent score should then be compared against a purpose-dependent benchmark to determine whether the model can be considered '*good*'.

• How these purpose-dependent benchmarks should be set is an open question to the hydrologic community.

**My Review Remarks:**

[1] I thoroughly enjoyed reading this Technical Note contribution by *Wouter, Knoben, Freer* and *Woods*, and I thank them for (re)raising some very important issues, and for their new/original contribution regarding the value that the KGE criterion takes on when using the mean flow as a benchmark. As such, I have no critique per se to offer regarding this paper, and compliment the authors on an excellent contribution to the literature.

Thank you for these kind words.

[2] Instead I would like to focus on some interesting points raised by this work. This review opportunity allows me to take the liberty of reminding the readers of some interesting points that were previously raised in *Schaefli and Gupta (2007)* and *Gupta et al (2009)*, that the authors allude to, *but which perhaps could be strengthened by the authors of the current work in their presentation*. Text between quotes is reproduced from the original papers.

We have made changes to the manuscript in order to strengthen our message like you suggest. Changes are detailed in response to your remaining comments.

[3] Beginning first with *Schaefli and Gupta (2007)*, that paper was about benchmarking. In it, we discussed the fact that the process by which anyone assesses and communicates model performance evaluation is of primary importance, and that "*the basic 'rule' is that every modelling result should be put into context, for example, by indicating the model performance using appropriate indicators, and by highlighting potential sources of uncertainty*".

We have added a sentence to the introduction to emphasize that benchmarks provide context for model performance (P3L1):

"However, using such a benchmark provides context for assessing model performance (*Schaefli and Gupta (2007)*)."

[4] We pointed out therein (as have others before and after us) that: a) the *"NSE value, while a convenient and normalized measure of model performance does not provide a reliable basis for comparing the results of different case studies"* b) the *"use of the mean observed value as a reference can be a very poor predictor (e.g. for strongly seasonal time series), or a relatively good predictor (e.g. for time series that are essentially fluctuations around a relatively constant mean value)"*. For example, *"In the case of strongly seasonal time series, a model that only explains the seasonality but fails to reproduce any smaller time scale fluctuations will report a good NSE value; for predictions at the daily time step, this (high) value will be misleading. In contrast, if the model is intended to simulate the fluctuations around a relatively constant mean value, it can only achieve high NSE values if it explains the small time-scale fluctuations"*.

We have added a sentence to the introduction to highlight the weakness of NSE of being not comparable between different flow regimes (P2L34, addition in bold):

"albeit this threshold could be considered a low level of predictive skill **(that is, it requires little understanding of the ongoing hydrologic processes to produce this benchmark); it is not an equally representative benchmark for different flow regimes (for example, the mean is not representative of very seasonal regimes but it is a good approximation of regimes without a strong seasonal component (Schaefli and Gupta, 2007));** and it is also a relatively arbitrary choice …"

[5] Therefore, the definition of an appropriate benchmark model is particularly important … to properly communicate how good a model really is, it is necessary to establish an appropriate reference (or benchmark model) for a given case study and a given modelling time step. In that paper we mention some examples, including: a) the interannual mean value for every calendar day proposed by *Garrick et al. (1978)* for systems having strong but relatively constant seasonality b) a simple adjusted precipitation benchmark (APB) where the rainfall is scaled to match the mean discharge and shifted in time by some optimum lag that reflects the time of concentration of the basin, and c) a smoothed version of the APB where a simple dispersion process (moving average) is added to adjust the smoothness of the scaled-down and translated precipitation to match the smoothness of the observed discharge, for example by maximizing the correlation between the adjusted precipitation and the observed flow *(Morin et al., 2002)*. Of course, many other possible benchmarks can be conceived, such as *"persistence"* (the next time steps' simulated flow is the same as the current time step's observed flow), some kind of linear or non-linear extrapolation into the future, and some kind of data-based time-series analytical model projection such as can be constructed by ARMAX or ANN methods.

We already provided references to some of these possible other benchmarks in our discussion (P5L14:"… but there is no direct reason to choose this benchmark over other options (see e.g. Ding, 2019; Schaefli and Gupta, 2007; Seibert, 2001; Seibert et al., 2018)."). Comparison of possible benchmarking options is not the focus of our paper and we have therefore chosen not to specifically mention what these alternatives are.

[6] In the conclusions to *Schaefli and Gupta (2007)*, we argued that the definition of an appropriate baseline for model performance, and in particular, for measures such as NSE (and by extension, KGE or any other model performance measure), should become part of the 'best practices' in hydrologic modelling, that *"Every modelling study should explain and justify the choice of benchmark"*, and that *"the benchmark should fulfill the basic requirement that every hydrologist can immediately understand its explanatory power for the given case study and, therefore, appreciate how much better the actual hydrologic model is"*.

This is a good and succinct description of why benchmarks are needed and we have quoted it verbatim in our revision (P5L17):

"As succinctly stated in Schaefli and Gupta (2007): *"Every modelling study should explain and justify the choice of benchmark [that] should fulfil the basic requirement that every hydrologist can immediately understand its explanatory power for the given case study and, therefore, appreciate how much better the actual hydrologic model is."*"

[7] Moving next to *Gupta et al (2009)*, we discussed the fact that the NSE, which is a dimensionless mathematical normalization of the mean squared error (MSE) criterion can be viewed as a classic skill score (*Murphy, 1988*), where 'skill' is interpreted as the comparative ability of a model with regards to a baseline 'model'. Further, as shown by *Murphy (1988)* and *Weglarczyk (1998)*, it is possible to decompose the NSE criterion into components (correlation, conditional bias, and unconditional bias) that facilitates a better understanding of what is causing a particular model performance to be 'good' or 'bad', while providing insight into possible trade-offs between the different components.

We have modified our introduction of the KGE metric to be clearer about its origin and also to include two concerns you mention further in this review, namely that KGE is not a perfect metric by itself and that other options could (should?) be explored (P3L3, additions in bold):

"The Kling-Gupta Efficiency (KGE, Eq. (2), Gupta et al., 2009) **is based on a decomposition of NSE into its constitutive components (correlation, variability bias and mean bias),** addresses several **perceived** shortcomings in NSE **(although there are still opportunities to improve the KGE metric and to explore alternative ways to quantify model performance)** and is increasingly used for model calibration and evaluation:"

[8] Our own particular diagnostic decomposition of NSE (and hence MSE) was developed in the context of our interest in hydrological modelling where, as we showed, interactions among these components (correlation, mean bias, and variance bias) can cause problems during model calibration – possibly leading to parameter estimates that are associated with large volume balance errors and/or underestimation of the variability in the flows. Further, we pointed out that many different combinations of the three components can result in the same overall value for NSE, leading to considerable ambiguity in the comparative evaluation of alternative model hypotheses.

We have added this notion about ambiguity of the overall NSE/KGE value to the text (P4L15, addition in bold):

"… see animated Figure S1 in Electronic Supplement 1 for a comparison of where NSE = 0 and KGE=1-√2 fall in the space described by KGE's r, a and b components for different CVs, **highlighting that many different combinations of r, a and b can result in the same overall NSE or KGE value**"

[9] Importantly, we also pointed out that, rather than trying to come up with a 'corrected' version of the NSE criterion, *the whole calibration problem can instead be viewed from the multi-objective perspective* (see e.g., *Gupta et al., 1998*), by focusing on the correlation, variability error and bias error as separate criteria to be optimized. *When we do so, if a compromise solution is desired, we can use the solution provided by the KGE or one of its alternatively weighted variants*.

We have added the mention of calibration as a multi-objective problem to the text (P7L20, additions in bold):

"… overall KGE score**, treating the calibration as a multi-objective problem (e.g. Gupta et al., 1998) with varying weights assigned to the three objectives**."

[10] We presented some comparative experimental results that show that when optimizing on KGE, there is a strong correlation between the values obtained for the KGE and NSE criteria, but when optimizing on NSE, the correlation between the values obtained for NSE and KGE is lower due to the fact that optimization on KGE strongly controls the values that the mean and variance ratio components can achieve, whereas optimization on NSE constrains these components only weakly. Overall, the use of KGE instead of NSE for model calibration tends to improve the bias and variability measures considerably while only slightly decreasing the correlation.

Our current Figure 1 shows that there is indeed some correlation between NSE and KGE values, but that there is large scatter in both directions. We have not changed the text in response to this comment.

[11] Finally, we pointed out that the NSE/MSE or KGE performance metric decomposition relates to the idea of diagnostic model evaluation, as proposed by *Gupta et al. (2008)*, *which is to move beyond aggregate measures of model performance that are primarily statistical in meaning, towards the use of (multiple) measures and signature plots that are selected for their ability to provide hydrological interpretation*. While the theoretical development behind the KGE provides one simple, statistically founded approach to the development of a strategy for diagnostic evaluation and calibration of a model, *we also pointed out that all other statistical properties beyond the mean and standard deviation (which are two long-term statistics of the data), such as timing of the peaks, and shapes of the rising limbs and the recessions of the hydrograph (i.e. autocorrelation structures), are lumped into the (linear) correlation coefficient as an aggregate measure*.

See below.

[12] We therefore suggested that a logical next step would be to consider other relevant diagnostic properties (such as for example, different aspects of flow timing and shape), but left those considerations are left for future work. For example, although not mentioned explicitly in *Gupta et al (2009)*, there is no reason that other (statistical or otherwise) aspects of model performance, such as "*skewness*", or "*particular quantiles*" etc., should not be integrated into the basis for model performance evaluation and, if desired, built into a "*KGE-like*" metric.

See below.

[13] However, the *explicitly stated purpose* of the *Gupta et al (2009)* study *was NOT to design an improved measure of model performance*, but instead: a) to show clearly that there are systematic problems inherent with any optimization that is based on mean squared errors (such as NSE), b) that "*the alternative criterion KGE was simply used for illustration purposes*" (many different alternative criteria would also be sensible), and c) that "*Ultimately the decision to accept or reject a model must be made by an expert hydrologist, where such a decision is best based in a multiple-criteria framework*", where tracking the mean bias, variance bias and correlation (and other possible) components can help.

See below.

**Concluding Remarks:**

[14] With this context, it would actually be useful for the community to strategically move beyond the use of single metrics for model performance assessment (and/or selection), whether NSE or KGE or any other that might be conceived, and to follow the spirit of *Gupta et al (2008)* by designing some reasonable and rational basis for selecting "*sets*" of metrics that provide meaningful diagnostic evaluation of a model.

We have combined the previous four remarks into a single new discussion paragraph (P8L8):

"However, aggregated performance metrics with a statistical nature, such as KGE, are not necessarily informative about model deficiencies from a hydrologic point of view (Gupta et al., 2008). In fact, while KGE improves upon the NSE metric in certain ways, Gupta et al. (2009) explicitly state that their intent with KGE was *"not to design an improved measure of model performance"* but only to use the metric to illustrate that there are inherent problems with mean-squared-error-based optimization approaches, They highlight an obvious weakness of the KGE metric, namely that many hydrologically relevant aspects of model performance (such as the shape of rising limbs and recessions, as well as timing of peak flows) are all lumped into the single correlation component. Future work could investigate alternative metrics that separate the correlation component of KGE into multiple, hydrologically meaningful, aspects. There is no reason to limit such a metric to only three components either and alternative metrics (or sets of metric components) can be used to expand the multi-objective optimization from three components to as many dimensions as are considered necessary or informative. Similarly, there is no reason to use aggregated metrics only and investigating model behaviour on the individual time-step level can provide increased insight in where models fail (e.g. Beven et al., 2014)."

[15] As pointed out by the current authors, to be meaningful, any such metrics should be accompanied by meaningful benchmarks. To be meaningful, these benchmarks should *not* be specified in an ad-hoc manner (such as NSE > 0.5 etc.) but should have some meaningful theoretical basis that conveys useful information to the decision maker.

See below.

[16] Indeed, I have often been contacted by researchers asking for some "*threshold*" values to use with KGE in their studies, and have always responded by discouraging such a practice and instead encouraging the use of the individual diagnostic components of KGE (and others that might be imagined) and setting associated thresholds using some meaningful basis.

See below.

[17] I do understand that, when performing studies involving large samples of data and/or many models, there is a tendency to want to use simple "*aggregate*" metrics in order to select or focus on a sub-set of "*good*" or "*poor*" models. However, there is arguably little to be gained by doing so by following the (arguably lazy) approach of using an aggregate metric that is not meaningfully interpretable.

See below.

[18] I sincerely hope that this current authored contribution will help to move the bulk of the community of hydrologic practitioners in the direction of using a more informative, and powerful, *diagnostic* (and necessarily multi-criteria) basis for model evaluation *that points to the nature of model deficiencies* and therefore to the modeling issues that need fixing.

See below.

[19] It might be helpful therefore, for the current authors to make some stronger arguments/comments in this direction, to encourage movement beyond the use of NSE and/or KGE, and thereby to a more powerful and robust approach to model assessment, as has been (slowly) pursued the case in some closely related communities (*Abramowitz 2012*).

We have combined the previous five remarks into a single rewrite of our concluding paragraph of the discussion section (P8L20):

"Regardless whether KGE or some other metric is used, the final step in any modelling exercise would then be comparing the obtained efficiency score against a certain benchmark that dictates which kind of model performance might be expected in this particular catchment (e.g. Seibert et al., 2018) and decide whether the model is truly skillful. These benchmarks should not be specified in an ad-hoc manner (e.g. our earlier example where the thresholds are set at NSE = 0.5 and KGE = 0.3 is decidedly poor practice) but should be based on hydrologically meaningful considerations. The explanatory power of the model should be obvious from the comparison of benchmark and model performance values (Schaefli and Gupta, 2007), such that the modeller can make an informed choice on whether to accept or reject the model, and make an assessment of the model's strengths and where current model deficiencies are present. Defining such benchmarks is not straightforward because it relies on the interplay between our current hydrologic understanding, the availability and quality of observations, the choice of model structure and parameter values, and modelling objectives. However, explicitly defining such well-informed benchmarks will allow more robust assessments of model performance (see for example Abramowitz, 2012, for a discussion of this process in the land-surface community). How to define a similar framework within hydrology is an open question to the hydrologic community."

We have added a final sentence to the conclusions that reflects the changes made above:

"More generally, a strong case can be made for moving away from ad-hoc use of aggregated efficiency metrics and towards a framework based on purpose-dependent efficiency metrics and benchmarks that allows for more robust model adequacy assessment."

We have added a final sentence to the abstract stating the same (P2L15, changes in bold):

"Therefore, we argue that modellers **who use the KGE metric** should not let their understanding of NSE values guide them in interpreting KGE values and instead develop new understanding based on the constitutive parts of the KGE metric and the explicit use of benchmark values to compare KGE scores against. **More generally, a strong case can be made for moving away from ad-hoc use of aggregated efficiency metrics and towards a framework based on purpose-dependent efficiency metrics and benchmarks that allows for more robust model adequacy assessment.**"

**Reviewer 2**

Summary:

The technical note provides interesting discussions on an interpretation of two metrics widely used in hydrologic community: NSE and KGE. First, the author reminds the readers that NSE is the metrics that quantify the performance compare to observed mean flow benchmark (NSE=0 indicates model performance is equivalent to this benchmark). The authors then state that there are many past studies that used KGE=0 as a threshold between bad and good model performance, same as NSE threshold. The authors point out KGE=0 does not hold the same meaning as NSE=0, and analytically show that KGE > -0.41 indicates that the model performs better than observed mean flow (if a modeler compares the model to mean flow using KGE). The authors made a direct comparison between NSE and KGE by random sampling of each KGE component and corresponding NSE, showing there is no unique relationship between two metrics, but their range of NSE value given a KGE partly depends on Coefficient of variation of the observed flow, indicating NSE and KGE cannot be directly compared. Finally, the authors that single, aggregated metrics like NSE and KGE might be misleading if the modeler looks for a specific model application (i.e., flood forecast need accuracy of high flow), and the modelers need to look more targeted metrics related to the application.

Comment:

I agree on all the major statements made in this technical note. I think one Figure presented in the note is unique contribution. It is similar to Fig 6d Gupta et al., 2009, but is expanded version and generated in the different context. I think this is very informative article, and great particularly for hydrologic practitioners who tend to quickly and intuitively evaluate the model with either NSE or KGE. I did not find any corrections/suggestions I can offer and therefor I recommend publish as is.

Thank you for your kind words. We appreciate you taking the time to read this manuscript and providing us with this review.

**Interactive comment by John Ding**

Equating the NSE and KGE scores

The authors raise an interesting question of whether or not the mean observed flow is an inherent benchmark of the NSE and KGE criteria. The mean flow is a base value intended by Nash and Sutcliffe (1970) to scale their NSE

5   score to between 0 and 1.  Corresponding KGE scores are -0.41 and 1 (Page 3, Line 10). Rescaling the KGE criterion to (KGE+0.41)/1.41 would produce a 0 to 1 scale.

*From our initial online response:*

*We agree with your comment that KGE can be rescaled so that the KGE score of the mean flow equals 0. Both Feyera et al (2018) and Towner et al (2019) use a generalized scaled KGE as a skill score metric [author's note:*

10   *our thanks to Shaun Harrigan for pointing this out]:*

$$KGE_{skill\ score} = \frac{KGE_{model} - KGE_{benchmark}}{1 - KGE_{benchmark}}$$

*This could potentially be of use for clearer communication of whether any model's KGE score exceeds the benchmark (i.e. all positive scores of KGE_{skill score}) or not (i.e. all negative scores on KGE_{skill score}).*

*However, scaling the KGE metric might introduce a different communication issue. In absolute terms, it seems*

15   *clear that improving on $KGE_{benchmark} = 0.99$ by using a model might be difficult: the "potential for model improvement over benchmark" is only 1-0.99 = 0.01. With a scaled metric, the "potential for model improvement over benchmark" always has range [0,1], but information about how large this potential was in the first place is lost and must be reported separately for proper context. If the benchmark is already very close to perfect simulation, a $KGE_{skill\ score}$ of 0.5 might indicate no real improvement in practical terms. In cases where the benchmark constitutes*

20   *a poor simulation, a $KGE_{skill\ score}$ of 0.5 might indicate a large improvement through using the model.*

*Similarly, scaling the metric might also reduce the ease of communication about model deficiencies. It is generally difficult to interpret any score above the benchmark score but below the perfect simulation (1) beyond 'higher is better', but an absolute KGE score can at least be interpreted in terms of deviation-from-perfect on its a, b and r components (assuming they are also reported). A score of KGE = 0.95 with r = 1, a = 1 and b = 1.05 indicates*

25   *simulations with 5% bias. A scaled KGE score of 0.95 cannot so readily be interpreted.*

*Therefore, we think that a scaled metric could be of use in some cases (the clear meaning of positive and negative values is useful) but also has some drawbacks: a scaled metric is not necessarily a more efficient way of communicating model performance (because still two values must be reported for proper context) and scaling also reduces the ease with which individual KGE components can be interpreted in terms of simulation deficiencies.*

30   *We will consider adding these thoughts to the discussion section in our manuscript.*

*We have added these considerations in a condensed way to the manuscript in a new section in the discussion (P5L25):*

**"3.3 On communicating model performance through skill scores**

*If the benchmark is explicitly chosen then a so-called skill score can be defined, which is the performance of any*

35   *model compared to the pre-defined benchmark (e.g. Hirpa et al., 2018; Towner et al., 2019):*

$$KGE_{skill\ score} = \frac{KGE_{model} - KGE_{benchmark}}{1 - KGE_{benchmark}}$$

The skill score is scaled such that positive values indicate a model that is better than the benchmark model and negative values indicate a model that is worse than the benchmark model. This has a clear benefit in communicating whether a model improves on a given benchmark or not with an intuitive threshold at $KGE_{skill\ score} = 0$, where negative values clearly indicate a model worse than the benchmark and positive values a model that outperforms the benchmark.

However, scaling the KGE metric might introduce a different communication issue. In absolute terms, it seems clear that improving on $KGE_{benchmark} = 0.99$ by using a model might be difficult: the "potential for model improvement over benchmark" is only 1-0.99 = 0.01. With a scaled metric, the "potential for model improvement over benchmark" always has range [0,1] but information about how large this potential was in the first place is lost and must be reported separately for proper context. If the benchmark is already very close to perfect simulation, a $KGE_{skill\ score}$ of 0.5 might indicate no real improvement in practical terms. In cases where the benchmark constitutes a poor simulation, a $KGE_{skill\ score}$ of 0.5 might indicate a large improvement through using the model. This issue applies to any metric that is converted to a skill score.

Similarly, a skill score reduces the ease of communication about model deficiencies. It is generally difficult to interpret any score above the benchmark score but below the perfect simulation (in case of the KGE metric, KGE = 1) beyond 'higher is better', but an absolute KGE score can at least be interpreted in terms of deviation-from-perfect on its a, b and r components. A score of KGE = 0.95 with r = 1, a = 1 and b = 1.05 indicates simulations with 5% bias. The scaled $KGE_{skill\ score} = 0.95$ cannot so readily be interpreted."

While worth searching for "a single perfect (hydrologic) model performance metric" (Page 4, Line 10), equally important if not more, in my opinion, is finding an alternate "starter" model to the mean flow one, the "no model" one in NSE. This will be a new benchmark or baseline against which the performances of other hydrologic models are to be measured. One of the "least skill(ful)" ones is a one–step linear extrapolation model of the observed flows. The predicted or forecast flow by extrapolation is: Qfore(t) =Qobs(t−1) + [Qobs(t−1)−Qobs(t−2)]. This is a simplest autoregressive model of order 2. It has been used on its own, i.e., outside the NSE, as a river forecast model. The NSE criterion may be modified by substituting the mean observed flow term, Qobs,in Equation (1), by the forecast flow. See Mizukami et al. (2019) cited by the authors for my previous comment on this (SC1 therein), the deficiency of the extrapolation model included.

Similar to our response to reviewer 1, we do not consider an overview of possible benchmark metrics within scope of this paper. We have therefore chosen not to explicitly mention autoregressive models in the text but have included a reference to this comment so that this may become part of future work on the topic (P5L14, addition in bold):

"… but there is no direct reason to choose this benchmark over other options (see e.g. **Ding, 2019**; Schaefli and Gupta, 2007; Seibert, 2001; Seibert et al., 2018)."

**References Cited:**

Abramowitz G, 2012, Towards a public, standardized, diagnostic benchmarking system for land surface models, Geoscientific Model Development, vol. 5, pp. 819 - 827, http://dx.doi.org/10.5194/gmd-5-819-2012

Ding, J.: Interactive comment on "Technical note: Inherent benchmark or not? Comparing Nash-Sutcliffe and Kling-Gupta efficiency scores" by Wouter J. M. Knoben et al., Hydrol. Earth Syst. Sci. Discuss., doi:https://doi.org/10.5194/hess-2019-327-SC1, 2019

Garrick M, Cumane C, Nash JE. 1978. A criterion of efficiency for rainfall-runoff models. Journal of Hydrology 36: 375–381.

Gupta HV, S Sorooshian and PO Yapo (1998), Towards Improved Calibration of Hydrologic Models: Multiple and Non-Commensurable Measures of Information, Water Resources Research, Vol. 34, No. 4, pp. 751-763

Gupta HV, T Wagener and YQ Liu (2008), Reconciling Theory with Observations: Towards a Diagnostic Approach to Model Evaluation, Hydrological Processes, Vol. 22 (18), pp. 3802-3813, DOI: 10.1002/hyp.6989.

Gupta HV, H Kling, KK Yilmaz and GF Martinez-Baquero (2009), Decomposition of the Mean Squared Error & NSE Performance Criteria: Implications for Improving Hydrological Modelling, Journal of Hydrology, Vol. 377, pp. 80-91, doi: 10.1016/j.jhydrol.2009.08.003.

Morin E, Georgakakos KP, Shamir U, Garti R, Enzel Y. 2002. Objective, observations-based, automatic estimation of the catchment response timescale. Water Resources Research 38: 1212, DOI: 10・1029/2001WR000808.

Murphy A (1988), Skill scores based on the mean square error and their relationships to the correlation coefficient. Monthly Weather Review 116: 2417-2424

Schaefli B and HV Gupta (2007), Do Nash values have value? Hydrological Processes, 21(15), 2075-2080, simultaneously published online as Invited Commentary in Hydrologic Processes (HP Today), Wiley InterScience, doi: 10.1002/hyp.6825

Weglarczyk S (1998), The interdependence and applicability of some statistical quality measures for hydrological models. Journal of Hydrology 206: 98-103